# Sparsity-Aware Grouped Reinforcement Learning for Designated Driver Dispatch

## Abstract

Designated driving service is a fast-growing market that provides drivers to transport customers in their own cars. The main technical challenge in this business is the design of driver dispatch due to slow driver movement and sparse orders. To address these challenges, this paper proposes Reinforcement Learning for Designated Driver Dispatch (RLD3). Our algorithm considers group-sharing structures and frequent rewards with heterogeneous costs to achieve a trade-off between heterogeneity, sparsity, and scalability. Additionally, our algorithm addresses long-term agent cross-effects through window-lasting policy ensembles. We also implement an environment simulator to train and evaluate our algorithm using real-world data. Extensive experiments demonstrate that our algorithm achieves superior performance compared to existing Deep Reinforcement Learning (DRL) and optimization methods.

## 1 Introduction

Designated driving, also known as chauffeur service and substitute driving, is an emerging business in the field of mobility service platforms. These platforms offer professional drivers to transport customers who are unable to drive, such as drunk drivers, rookie drivers, and tired drivers. The designated driver arrives with an electric scooter and drives the customer to their destination, as shown in Figure 1. The platform controller manages dispatching behaviors to improve customers' experience and drivers' income. Designated driving has become a significant and promising industry, with a market size of over $4$ billion in China (BusinessGrowthReport, 2022).

One of the critical challenges in this industry is the design of driver dispatch, also known as the fleet management problem. While typical ride-hailing platforms focus on improving the matching quality between drivers and customers, designated driving platforms still struggle to find a driver for each order. This is due to the sparsity of designated drivers and their slow movement. Besides, designated orders have "hub-and-spoke" structures, with origins concentrated in specific hotspots (e.g., bars, restaurants) and destinations primarily being residential areas, which often result in drivers being far away from potential customers.

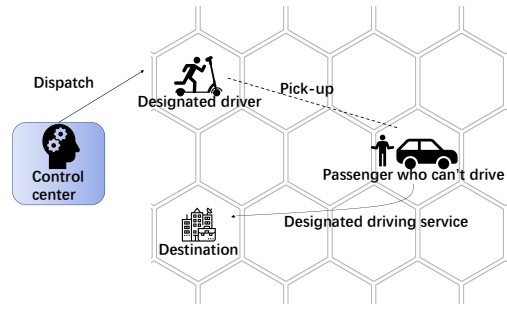

Figure 1: Designated driving.

Optimization methods are commonly used to address fleet management problems (Zhang et al., 2017; Robbennolt & Levin, 2023), but they require a certain level of modeling for the supply and demand dynamics, which is complex in the real world. Recently, many Deep Reinforcement Learning (DRL) approaches have been proposed to solve fleet management problems in ride-hailing services (Oda & Joe-Wong, 2018; Al-Kanj et al., 2020; Zhang et al., 2020; Liu et al., 2020; Shou & Di, 2020; Qin et al., 2021; Eshkevari et al., 2022; Liu et al., 2022; Zheng et al., 2022). However, designated drivers present unique challenges compared to traditional taxi ride-hailing systems. The challenges stem mainly from the sparsity, which can be attributed to three key factors. Firstly,

the dataset itself exhibits sparsity. In the case of designated driving, the number of drivers is considerably smaller compared to taxi drivers, resulting in a sparser spatial-temporal distribution. To illustrate, our dataset collected from Hangzhou, a Chinese city with a population of approximately 10 million, has only around 3,000 designated drivers and nearly 13,000 order requests per day. Secondly, individual drivers experience sparse feedback on the direct matching of orders. As designated drivers move slowly and are often located far away from available orders, each driver, on average, completes only 3 to 4 orders per day. Additionally, after matching with an order, the driver also spends a significant amount of time on the way to pick up the client. Thirdly, the cross-effect of agents is sparse and long-lasting. This is due to the slow and continuous impact of driver movements on their distribution, which is crucial in fleet management. Before each driver is matched with an order, they typically engage in continuous movement for several quarters. Therefore, considering the lasting impact becomes more crucial than focusing solely on the transient movements of other agents. Moreover, the heterogeneity and scalability of agents pose additional challenges for traditional MARL algorithms. Factors such as varying speeds and mileage limitations among different drivers, as well as the fluctuating number of drivers commuting to work each day, further contribute to these challenges.

To address these challenges, this paper proposes a group-sharing window-lasting Reinforcement Learning framework for Designated Driver Dispatch problems, RLD3. We model the problem as a Decentralized Partially Observed Markov Decision Process (Dec-POMDP), capturing the fact that drivers usually have local observations. RLD3 incorporates several novel designs. Firstly, we introduce a group-sharing structure, where agents are classified into several groups. Agents within the same group share the same network parameters and experience data. This design strikes a balance among sparsity, heterogeneity, and scalability. Secondly, we design a reward structure for the DRL algorithm. This specially designed reward estimates the potential of the neighborhood around the driver by considering the distances of all unmatched orders in that area, addressing the issue of sparse feedback. It also incorporates complicated movement constraints by applying heterogeneous moving costs. Thirdly, we design a time window to calculate the cumulative actions of agents during consecutive execution periods, allowing estimation of other agents' policies and making it suitable for sparse and lasting multi-agent interactions. Finally, we implement an environment simulator using real-world designated driving datasets and conduct extensive experiments to train and evaluate different algorithms. The results demonstrate that RLD3 outperforms existing DRL benchmarks and optimization policies in terms of completed order numbers and adherence to moving constraints.

The main contributions of this paper are summarized as follows:

i) We are the first to formulate a general Dec-POMDP framework for designated driver dispatch problems in designated driving markets.

ii) We propose a novel MARL algorithm, RLD3, to address the challenges of designated driver dispatch and achieve trade-off among scalability, heterogeneity, and sparsity. This algorithm builds upon group-sharing structures and window-lasting agent interactions with a potential/cost-aware reward.

iii) We design a designated driving simulator using real-world datasets and conduct extensive experiments. The results show that RLD3 efficiently learns system dynamics and outperforms existing DRL and optimization methods.

## 2 RELATED WORK

**Driver Dispatch.** As mentioned in Section 1, the driver dispatch problem has been extensively investigated in the existing literature. Two prominent methodologies have garnered significant attention: optimization algorithms (Zhang et al., 2016; Robbennolt & Levin, 2023) and DRL-based algorithms (Oda & Joe-Wong, 2018; Al-Kanj et al., 2020; Zhang et al., 2020; Liu et al., 2020; Shou & Di, 2020; Qin et al., 2021; Eshkevari et al., 2022; Liu et al., 2022; Zheng et al., 2022). Optimization algorithms leverage historical driver and order distributions to formulate dispatch policies, but they require precise knowledge of demand-supply dynamics, which is challenging to obtain in the real world. DRL-based algorithms are powerful in solving driver dispatch problems as they can learn a parametric model without relying on strong problem-based assumptions and can optimize long-term effects through sequential decision-making. However, taxi drivers move at a faster speed,

and taxi orders are much denser and more balanced. These features significantly reduce the sparsity challenges faced by traditional DRL-based dispatch algorithms. Thus, it is difficult to directly transfer the models and algorithms to the designated driving platform.

**Reinforcement Learning.** Reinforcement learning (RL) techniques have shown promise in addressing complex multi-agent problems. The Multi-Agent Deep Deterministic Policy Gradient algorithm (MADDPG) (Lowe et al., 2017) extends the Deep Deterministic Policy Gradient (DDPG) (Lillicrap et al., 2016) and Deterministic Policy Gradient algorithms (Silver et al., 2014) by using deep neural networks to approximate action values and handle agent interactions. Such algorithms within the traditional CTDE paradigm Claus & Boutilier (1998) often allow agents to achieve good overall performance by utilizing heterogeneous strategies. However, due to the independent nature of each agent's policy, they encounter the challenge of sparse feedback in the designated driving problem, leading to lower efficiency in exploration and policy learning.

To address sparsity, Random Network Distillation (RND) (Burda et al., 2019) uses an additional value function to estimate intrinsic reward in order to enhance exploration. In the designated driving platform, due to the unique "hub-and-spoke" structure of orders, the hotspots of orders are more concentrated. Exploring non-semantic information would result in excessive driver movement costs. Curriculum Learning approaches, such as Curriculum Deep Reinforcement Learning (Hacohen & Weinshall, 2019) and Relevant Curriculum Reinforcement Learning (Flet-Berliac & Preux, 2020), help in learning from sparse feedback by planning the neural network's learning path. However, planning learning paths in multi-agent scenarios is challenging due to the complex dynamics of cooperation and competition among drivers. Mean-Field Reinforcement Learning (MFRL) techniques, such as Mean Field Multi-Agent Reinforcement Learning (MFMARL) (Yang et al., 2018) and Multi-Agent Mean Field Q-Learning (Ganapathi Subramanian et al., 2020), model agent interactions as the interaction between a single agent and a field effect. Mean-field methods can address the issue of sparse agent distributions but lack consideration for the lasting interaction of different drivers, which should be taken into account since designated drivers have slow movement and complex constraints.

To address scalability and heterogeneity, Hierarchical Reinforcement Learning (HRL) approaches, such as Feudal HRL (Vezhnevets et al., 2017), Data-Efficient HRL (Nachum et al., 2018), and Model-Free HRL (Rafati & Noelle, 2019), decompose large-scale problems into sub-agents. But in the context of designated driver dispatch, additional attention should be paid to the complex interactions among agents and various sparsity issues as mentioned before.

## 3  RLD3: REINFORCEMENT LEARNING FOR DESIGNATED DRIVER DISPATCH

In this section, we present the formulation of the Decentralized Partially Observed Markov Decision Process (Dec-POMDP) for the designated driver dispatch problem. We introduce three unique designs in our algorithm: the grouped structure, the potential reward, and the lasting agent interaction.

### 3.1  FORMULATION

We consider the designated driving service in one metropolis. Each day, there are $N$ drivers with random initialization. Orders appear in the system at specific times and locations. Unmatched orders have limited patience and will be canceled after a waiting period following a Poisson distribution. Drivers that have completed their corresponding orders leave the system after off-duty time.

For simplicity, we assume that time in the system is slotted, with each time step corresponding to 30 seconds. At each time step, the platform decides the dispatch movement for every idling driver. We assume that drivers fully comply with movement instructions. The statuses of drivers and orders are updated until the next time step due to matches between idling drivers and unmatched orders, as well as the generation/completion processes.

The Dec-POMDP formulation $\langle N, \mathbb{S}, \mathbb{O}, \mathbb{A}, \mathcal{P}, \mathbb{R}, \gamma \rangle$ is presented as follows:

**Agent**  $i \in [N]$: Each driver is considered an agent, resulting in a total of $N$ unique agents. The platform can only dispatch idling drivers, as each agent can be in one of three statuses: offline, idle, or serving orders at any given time $t$.

**State**   $s \in \mathbb{S}$: At each time $t$, a global state is maintained, taking into account the status of all drivers and orders. This includes coordinates, moving distance, working status, serving targets, and moving targets for drivers. The state also includes calling time, patience, origin, destination, and serving status for orders.

**Observation**   $s \mapsto_i o_i \in \mathbb{O}$: Drivers have partial observations of the state $s$. In our implementation, each agent's observation is represented by a 22-dimensional vector:

$$([\#\boldsymbol{order}], [\#\boldsymbol{driver}], [\mathbf{min}\, \boldsymbol{dist}], t, lat, lng, move), \tag{1}$$

where the first three terms denote the number of orders to be matched, the number of idling drivers, and the distance to the closest order in six-segment-direction neighborhoods as shown in Figure 12. The last four terms represent time, latitude, longitude, and the distance the driver has already moved.

**Action**   $a_1 \times \cdots \times a_N \in \mathbb{A}$: The platform proposes a joint action instructing the movement policy for all available drivers based on their observations $o^t$ at time $t$. The action space for an individual agent consists of seven discrete actions including six neighboring directions and staying at the current location as shown in Figure 11. Agents located at the boundary and corners have a smaller action space.

**State Transition**   $\mathcal{P} : s \times \boldsymbol{a}_{[N]} \mapsto s'$: The movement of drivers, along with order updates and matches between drivers and orders, induces state transitions in the environment.

**Reward**   $r_i \in \mathbb{R}$: After executing an action, each agent receives its distinct instant reward $r_i$. The instant reward $r_t^i$ is defined as the sum of the immediate match reward, neighborhood potential reward, and move cost:

$$r_i^t = mt_i^t + nb_i^t + mv_i^t. \tag{2}$$

Immediate match reward $mt_i^t$ directly relates to the gross merchandise volume of the platform, which is the objective of our algorithm. To optimize volume without using discriminatory personal information, the immediate match reward is set to a fixed number:

$$mt_i^t = \begin{cases} 50, & \text{if agent } i \text{ is matched with an order at } t; \\ 0, & \text{otherwise.} \end{cases} \tag{3}$$

The move cost and neighborhood potential reward will be introduced in Section 3.2 and 3.3.

## 3.2   TOWARDS DATASET SPARSITY THROUGH GROUP SHARING

We introduce the concept of group sharing to address dataset sparsity issues in our approach. Meanwhile, we estimate the influence between these groups using the mean-field effect to ensure heterogeneity and scalability.

In real-world scenarios, drivers can be classified into several types based on their cost conditions. These endogenously heterogeneous agents are naturally mediated into several groups. Agents within the group share the same network along with their experience data in the training process. Specifically, we divide the $N$ agents into $M$ classes, where $M$ is a fixed number.

To control grouped drivers' moving distance, we include move cost $mv_i^t$ as a regularizer that influences the behavior of agents in the reward. The move cost for agent $i$ at time $t$ is set as follows:

$$mv_i^t = \begin{cases} -c_j, & \text{if agent } i \text{ moves;} \\ 0, & \text{if agent } i \text{ stays;} \end{cases} \tag{4}$$

where $j$ is the group index of agent $i$.

RLD3 utilizes double critic-networks and double actor-networks, with the delayed copy used for soft-update. During the training stage, a group network can access the experienced data of all agents belonging to that group, stored in a replay buffer. Therefore, a network can efficiently explore different individuals of the same category in the metropolis and gather more experiences. During the execution stage, each agent calls its corresponding group network to perform policy execution independently. The policy input for each agent is based on its current observation while the output is its

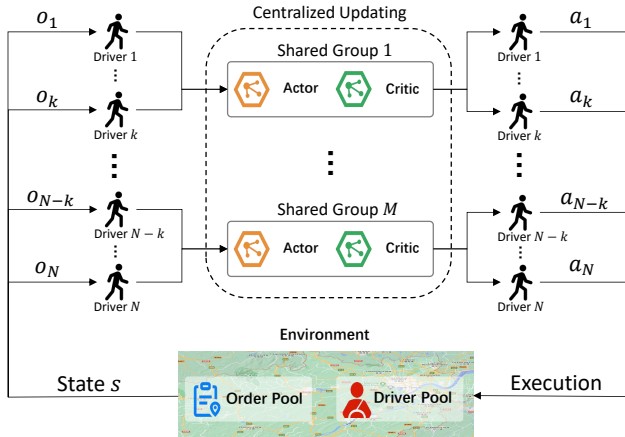

Figure 2: Information flow in the execution stage.

deterministic action. To transform the continuous output seven-dimensional vector into a deterministic action, the last layer uses Gumble-Softmax (Jang et al., 2017). Such mixed strategy ensures that even agents of the same group at the same location may execute different discrete actions, avoiding competition among agents. The information flow during the execution stage is illustrated in Figure 2.

### 3.3 Towards Feedback Sparsity through Space Potential

Since the immediate match reward is highly sparse for the DRL method in designated driving platforms (i.e., it only occurs at the time step with a successful order match, which is rare), we introduce a dense neighborhood potential reward $nb_i^t$ to reflect the potential value of the current area. The intuition is that the distance to an order in the neighborhood reflects how fast an agent can pick up the order. Almost all orders in the neighborhood are attractive to the driver, although the closest ones are especially attractive.

Specifically, we assign potential values to nearby unmatched orders, with higher feedback given to closer orders. We then sum up all potential values to represent the total potential value of the driver's current position. This provides reward feedback to the driver at every time step, compensating for the sparse immediate match reward. The potential reward is defined as follows:

$$nb_i^t = (d^* + 0.1)^{-0.5} + 0.1 \times \sum_{\text{neighbor order } j} (d_{ij} + 0.1)^{-0.5}, \tag{5}$$

where $d_{ij}$ denotes the distance from driver $i$ to order $j$, and $d^*$ denotes the distance to the closest order. The power index is set to $-0.5$ to ensure that the potential reward increases as the distance approaches and is a convex function, in order to encourage designated drivers to approach a specific order rather than maintain an equal distance from all orders.

### 3.4 Towards Interaction Sparsity through Window Lasting

In designated driving platforms, agents are often far away from each other, resulting in sparse and long-term agent interactions instead of single-step actions. For example, a driver's income is not directly influenced by the short-term actions of drivers located far away, but rather by the accumulated distribution changes caused by the lasting movements of drivers. Therefore, we use the average action over a time window, instead of a single-step action, when considering other agents' policies.

To achieve this, in addition to recording regular tuples $(s, \boldsymbol{a}_{[N]}, \boldsymbol{r}_{[N]}, s')$, the buffer calculates and stores the window-lasting actions for all agents. The window-lasting action $\hat{a}_i$ represents the average

of sequential idling actions for the last $W$ time steps:

$$\hat{a}_i^t = \mathbb{E}\left[a_i^s\right], s \sim [t - W, t] \cap T_{\text{last idle}}, \tag{6}$$

where $T_{\text{last idle}}$ refers to the most recent period in which the driver was idling, considering possible different idling periods that may result in diverse moving directions. Thus, the mean-field effect for group $j$ is defined as:

$$g_j^t = \mathbb{E}_{i \in \text{group } j}\left[\hat{a}_i^t\right]. \tag{7}$$

Additionally, we use an encoder in the input of the critic to handle complex state representations and their varying dimensions. This encoder is responsible for the distribution of the current unmatched orders and idling agents respectively. We employ the K-Means algorithm (Hartigan & Wong, 1979) for this encoder. Therefore, the input structure of the critic network is $Q_i(o_i, encode(s), a_i, \boldsymbol{g}_{[M]})$, as shown in Figure 3. All networks utilize two fully connected layers and the GELU activation function (Hendrycks & Gimpel, 2016).

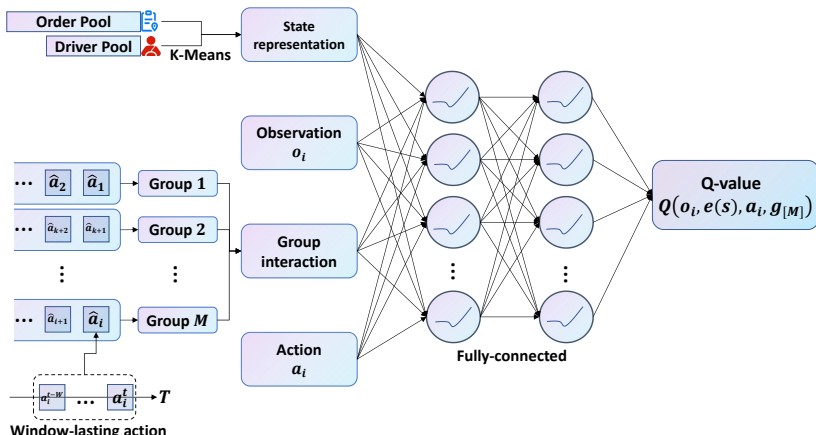

Figure 3: Network structure.

## 3.5 NETWORK UPDATE

The network update follows the gradient-based actor-critic paradigm. To ensure smoother driver trajectories, we add the temporal difference of adjacent actions $H(a, a') = \|a - a'\|_2$ to the Bellman loss as the critic loss. After incorporating the above techniques, the loss function for the value network becomes:

$$\begin{aligned}
\mathcal{L}(\theta_i) =& \mathbb{E}_{sample_i^t}\left[\left(Q_i^\pi\left(o_i, encode(s), a_i, \boldsymbol{g}_{[M]}\right) - y\right)^2 + \lambda H(a_i, a_i')\right], \\
y =& r_i + \gamma Q_i^{\pi'}\left(o_i', encode(s'), a_i', \boldsymbol{g}_{[M]}'\right).
\end{aligned} \tag{8}$$

Similarly, the gradient of the policy network is now:

$$\nabla_{\theta_i} J(\pi_i) = \mathbb{E}_{sample_i^t}\left[\nabla_{\theta_i}\pi_i(a_i \mid o_i)\nabla_{a_i}Q_i^\pi\left(o_i, encode(s), a_i, \boldsymbol{g}_{[M]}\right)\big|_{a_i=\pi_i(o_i)}\right]. \tag{9}$$

The complete algorithm framework is summarized in Algorithm 1.

## 4 SIMULATOR & EXPERIMENT

We design and implement a simulator based on real-world datasets to train and evaluate RL algorithms for the designated driver dispatch problem. We then conduct experiments on our proposed model using the simulator and real-world data. We sample 50 drivers and 500 orders for the training stage. Each experiment is repeated with 4 different seeds, and the average results with confidence intervals are presented. To mitigate the sparsity issue in early training, we use the first 100 episodes for random exploration.

---

**Algorithm 1** RLD3.

---

**Require:** order data, driver pool $[N]$, episode number $MAX$, episode length $T$, learning rate $\lambda$, update rate $\tau$, batch size $S$, group number $M$, window size $W$.
1: **for** episode from 1 to $MAX$ **do**
2:     Initialize environment and receive an initial state $s$.
3:     **for** $t$ from 1 to $T$ and not all drivers are off-line **do**
4:         Generate action $a_i = \pi_i(o_i)$.
5:         Execute action $(a_1, a_2, \cdots, a_N)$ and observe reward $\boldsymbol{r}$ and next state $s'$.
6:         Push $(s, \boldsymbol{a}, \boldsymbol{r}, s', \hat{\boldsymbol{a}})$ into buffer.
7:         $s = s'$.
8:         **for** group $j$ from 1 to $M$ **do**
9:             Sample a batch of $S$ samples $(s, o_i, a_i, r_i, s', \hat{\boldsymbol{a}})(i \in$ group $j)$ from replay buffer.
10:            Update critic by minimizing $\mathcal{L}(\theta_j)$.
11:            Update actor using sample policy gradient $\nabla_{\theta_j} J$.
12:         **end for**
13:         Update the target network parameter for each agent $i$ by $\theta_i' = \tau\theta_i + (1 - \tau)\theta_i'$.
14:     **end for**
15: **end for**

---

### 4.1 SIMULATOR

The simulator is built based on real-world designated driver and order datasets from Hangzhou, a city in China. The datasets include over 3,000 drivers and nearly 13,000 orders per day. Each order's information consists of its coordinates and the time of generation, match, completion, and possible cancellation. Each driver's information includes their online time, offline time, and online coordinates. The simulator models the entire process of how the states of drivers and orders evolve. It includes a driver dispatch module that allows for the repositioning of any idling driver. The simulator serves as a training environment for RL algorithms and can also evaluate the performance of various dispatch policies. The detailed introduction of the simulator is in Appendix A.

### 4.2 PERFORMANCE COMPARISON

We compare the performances of our algorithm with existing DRL methods and optimization-based policies. The benchmark DRL algorithms include independent DDPG (Lillicrap et al., 2016), MADDPG (Lowe et al., 2017), MAMFRL (Yang et al., 2018), and multi-agent version RND (Burda et al., 2019). These algorithms are applied with the immediate match reward and move cost to achieve a trade-off between match and movement. All DRL algorithms use the same two hidden layers of dimension 64 and batch size of 512. The update rate is set to 0.01, and the learning rate policy uses the Adam optimizer (Kingma & Ba, 2015) with an initial rate of 0.01. All DRL algorithms are trained for 1000 episodes. In RLD3, the lasting window size is set to 60 steps, and the group number is set to 5.

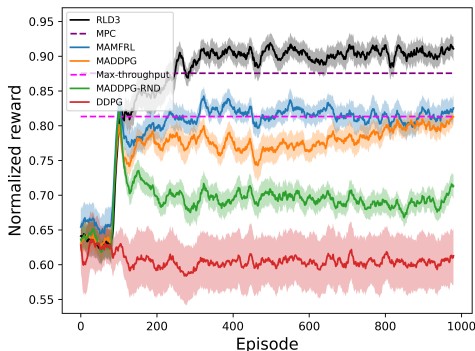

Figure 4: Training performance. The order of the legends in the figure is the same as the order of performances in the last episode.

The optimization-based policies included order-oriented random-walk, Max-throughput dispatch policy (Robbennolt & Levin, 2023), and model predictive control (MPC) (Zhang et al., 2016). To ensure fairness in comparison, optimization-based methods estimate the current order and driver dynamics based on past history.

Table 1: Testing performance. The testing performance is evaluated based on the model that has the best episodic performance, while the IID generalization performance is measured using an additional testing dataset of 10 episodes that are separated from the training dataset.

| | Algorithm | Testing performance | | IID generalization | |
|---|---|---|---|---|---|
| | | Order | Distance (km) | Order | Distance (km) |
| Our Algorithm | RLD3 | $237.2 \pm 3.4$ | $7.0 \pm 1.3$ | $234.0 \pm 3.9$ | $7.0 \pm 1.3$ |
| Taxi-dispatch | Deep-dispatch | $233.3 \pm 2.3$ | $15.5 \pm 2.4$ | $229.0 \pm 2.3$ | $15.1 \pm 2.3$ |
| DRL-based | DDPG | $186.9 \pm 5.2$ | $27.8 \pm 3.0$ | $183.1 \pm 5.5$ | $27.9 \pm 3.1$ |
| | MADDPG | $215.7 \pm 3.7$ | $29.6 \pm 0.6$ | $212.0 \pm 4.4$ | $30.2 \pm 0.5$ |
| | MADDPG-RND | $228.6 \pm 3.5$ | $65.3 \pm 0.7$ | $224.0 \pm 3.7$ | $66.3 \pm 0.9$ |
| | MAMFRL | $224.3 \pm 3.7$ | $34.3 \pm 5.3$ | $221.1 \pm 4.8$ | $34.9 \pm 5.5$ |
| Optimization | Random | $180.1$ | $35.3$ | $178.3$ | $34.4$ |
| | Max-throughput | $229.8$ | $73.2$ | $228.8$ | $73.1$ |
| | MPC | $228.1$ | $1.7$ | $228.2$ | $1.5$ |

As shown in Figure 4 and Table 1, our model outperforms all other algorithms in terms of the number of completed orders and had a smaller moving distance compared to methods that had similar completed order performance. As mentioned in Section 2, RND falls into no-semantic exploration due to always moving; MADDPG and MAMFRL fail to differentiate the value of different directions when there are no nearby orders, resulting in a significant amount of random walking. For optimization baselines, the Max-throughput policy optimizes the Lyapunov drift by treating the drivers as servers, which in turn leads to intense competition among drivers for orders. As one of the most popular algorithms in control theory, MPC outperforms the DRL baselines, except for our proposed algorithm RLD3.

## 4.3 INDEPENDENT AND IDENTICALLY DISTRIBUTED (IID) GENERALIZATION

We conducted IID Generalization experiments to assess the robustness and generalization of our algorithm. In IID Generalization, it is assumed that the data points in both the training and testing datasets are drawn independently and identically from the same underlying distribution (Kirk et al., 2023). The generalization performance is then synonymous with the test-time performance from IID samples. We sampled another 500 orders from the real-world data that were not seen during training in every episode.

As shown in Table 1, our algorithm did not decline significantly in IID performance and still outperformed other methods. An interesting phenomenon is that all algorithms demonstrate good IID generalization performance. This is because the designated driving platform itself exhibits sparsity, and the hotspots of orders are concentrated. Since we maintain the same initial state for all drivers and the same order underlying distribution in the IID generalization test, drivers are still able to effectively transfer the learned hotspot information from previous experiences when moving.

## 4.4 ABLATION STUDY

We conducted an ablation study on the group-sharing structure, agent interaction design, state encoder, and reward design to gain insights into our model's settings and behavior.

**Group Number.** The group number is a typical hyperparameter that determines the number of agent types. A larger group number can better represent the heterogeneity of drivers, but it also increases the storage pressure and training time. Additionally, a large group number may not learn well in sparse feedback situations. The results in Table 2 show that the group-sharing structure helps improve the performance of MADDPG and our proposed algorithm RLD3.

**Window-lasting Agent Interaction.** Our algorithm uses a window-lasting policy ensemble in the updating stage to better learn the cross-effects of other agents' policies. We evaluated the algorithm without the window average.

As shown in Table 2, the model without the window-lasting interaction cannot learn others' policies well. This could be due to high-frequency fluctuations in agent actions that are difficult to learn, as

Table 2: Ablation study.

| Algorithm | Order | Distance (km) |
|---|---|---|
| RLD3 | $237.2 \pm 3.4$ | $7.0 \pm 1.3$ |
| RLD3 for 1 group | $150.5 \pm 9.2$ | $21.5 \pm 1.2$ |
| RLD3 for 50 groups | $231.7 \pm 3.6$ | $9.1 \pm 0.4$ |
| MADDPG for 5 groups | $223.0 \pm 3.6$ | $24.5 \pm 1.2$ |
| MADDPG-RND for 5 groups | $211.5 \pm 7.4$ | $56.1 \pm 1.8$ |
| MAMFRL for 5 groups | $227.0 \pm 10.2$ | $6.1 \pm 1.7$ |
| RLD3 without window-lasting | $229.5 \pm 3.2$ | $27.1 \pm 3.4$ |
| RLD3 without state encoder | $232.2 \pm 3.7$ | $27.2 \pm 1.6$ |
| RLD3 without potential reward | $223.8 \pm 3.8$ | $6.7 \pm 1.2$ |
| RLD3 without move cost | $210.7 \pm 6.7$ | $48.3 \pm 2.6$ |

well as the fact that single-step actions may not be executed for agents that are not idling. Consequently, the value function underfits when other agents' policies are ensembled without the window average.

**State Encoder.** To capture the distribution information of orders and drivers during the training stage, we employ an encoder to encode the system's state. It is worth noting that due to the varying number of orders and drivers, the dimensions of the state vector are constantly changing, making it difficult to directly utilize by the value function. Therefore, we extract the distribution information of orders and drivers separately using the K-Means method.

As shown in Table 2, such a state encoder can assist DRL algorithms in better understanding the state of the designated driving platform, particularly in extracting driver-order distribution information. Additionally, when comparing the performance of our algorithm without the state encoder and traditional DRL baselines that only utilize observation information, our algorithm still outperforms them due to the benefits of group-sharing and window-lasting interaction techniques.

**Reward Design.** We compared different reward components by removing the neighborhood potential reward and move cost, as shown in Table 2. All reward settings were tested with our proposed group-sharing structure and training process. The dense potential reward not only increases performance but also stabilizes the training process, as indicated by the much smaller value function loss. While the model without the cost falls into a suboptimal situation where only order numbers are optimized, ignoring distance constraints.

## 5 CONCLUSION

In this paper, we addressed the problem of driver dispatch in designated driving platforms, which is a complex scenario with sparsity issues and strict constraints. To capture the spatiotemporal dynamics of imbalanced demand-supply relations, we proposed a novel multi-agent deep reinforcement learning (DRL) algorithm based on the decentralized partially observed Markov decision process (Dec-POMDP) formulation. Our algorithm leverages a group-sharing structure and a specially designed reward to address the trade-off between sparsity, scalability, and heterogeneity. The window-lasting agent interaction technique enables our algorithm to handle the long-lasting cross-effect of agents.

Through extensive experiments on a simulator based on real-world data, we demonstrated that our algorithm outperformed traditional optimization-based policies and existing DRL algorithms in terms of completed order numbers and moving constraints. The results highlight the effectiveness of our approach in addressing the challenges of the designated driver dispatch problem.

In future work, we aim to make the grouping process trainable by incorporating self-supervised algorithms such as clustering. This would enable us to better model the interactions between agents and enhance the performance of our algorithm. Additionally, we are interested in studying the impact of non-compliance on the performance of driver dispatch, as existing literature often assumes drivers' full compliance. Understanding and addressing non-compliance issues can further enhance the effectiveness of our algorithm in real-world scenarios.

ETHICS STATEMENT

During the data collection process, we filtered out all personal information regarding designated drivers and orders and used virtual IDs to prevent the leakage of behavior patterns. In the experimental design, we did not employ any discriminatory strategies towards any specific driver or order. Our optimization objective is to maximize the gross merchandise volume of the entire platform, thereby improving service quality while increasing workers' income.

REPRODUCIBILITY STATEMENT

To facilitate reproducibility, we provide a detailed description of the models and training details in the main text. We also list all relevant parameters in the appendix. If the paper is accepted, we will provide an open-source link in the camera-ready version.

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

## APPENDIX

This appendix contains 3 sections. The first section provides details about the designated driving simulator along with the real-world dataset. The detailed hyperparameters and random seeds are presented in the second section. The third section presents additional experimental results to analyze the differences between the baseline and our algorithm. This analysis will assist us in better utilizing different DRL algorithms in other applications.

## A  SIMULATOR

As introduced in Sec 4.1, we have designed and implemented a designated driving simulator based on real-world datasets to train and evaluate RL algorithms for the designated driver dispatch problem.

### A.1  REAL-WORLD DATASET

The simulator is built on the data from a designated driving platform in Hangzhou, a Chinese city with a population of tens of millions. It includes over 3,000 drivers and nearly 13,000 orders per day. Each order's information consists of the coordinates and the time of its generation, match, completion, and possible cancellation. Each driver's information consists of online time, offline time, and online coordinates. The data collection process does not include personal information about drivers and orders. To prevent the leakage of driver or passenger behavior patterns, we also utilize virtual IDs.

### A.2  ORDER STATE TRANSFER

For the order process, every appearing order enters the order pool and waits to be matched at a predetermined real-world generation time. During the waiting period, if an unmatched order is not answered within a specified period (15 minutes), it enters the timeout state and fails. Additionally, each order may be canceled via a Poisson Process with a mean patience of 8 minutes if the order is

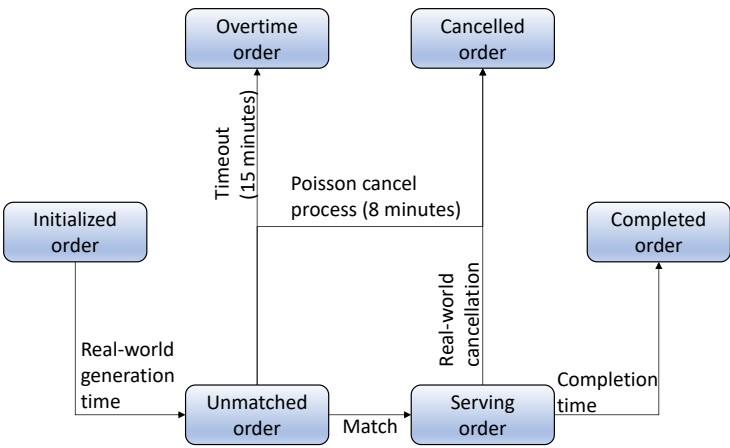

Figure 5: Order state-transfer.

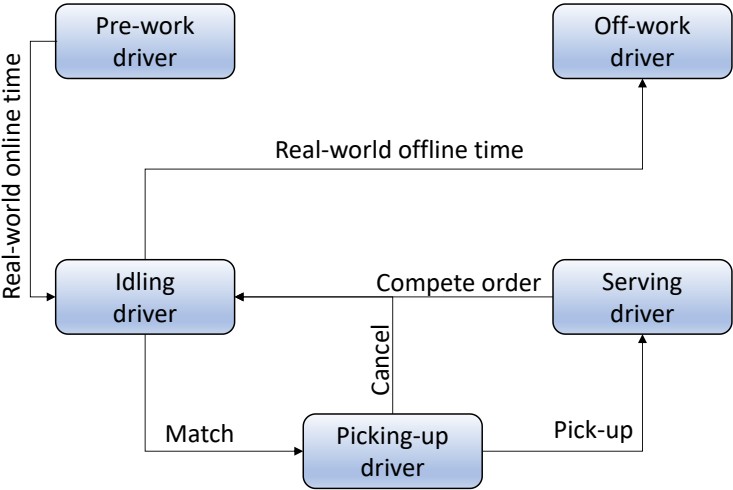

Figure 6: Driver state-transfer.

not canceled due to a lack of patience in the real world. This process is memoryless and independent for each order. However, if an order is canceled by the customer in the real world, the patience of the order will be set to the actual value. When an unmatched order is matched with a driver, it enters the on-service state and then transitions to the completion state after the expected completion time. The whole state transfer for orders is illustrated in Figure 5.

## A.3 DRIVER STATE TRANSFER AND DISPATCH

Each driver has a scheduled on-work and off-work time. When the current simulation time exceeds the online time of a driver in the driver pool, the driver enters the idle state from its actual location in the real world. Idling drivers can either move to a given location according to the dispatch policy or match with an order for service. When an idling driver is matched with an order, the driver immediately moves to pick up the customer. Once the order is completed, the driver returns to the idle status until the simulation time exceeds their offline time. The whole state transfer for drivers is illustrated in Figure 6.

### A.4 MATCHING MODULE

The simulator applies a simple two-step driver-searching algorithm to match drivers and nearby orders. This algorithm called the AB-circle algorithm, is intuitive and is used by the platform from which the dataset is provided. At each time step, for orders around which there are idling drivers within the A-circle (with a radius of 3000 meters), the algorithm assigns the order to the closest driver to minimize pick-up time. After matching all such orders, the algorithm calculates a global optimal match between orders and drivers within the B-circle (with a radius of 5000 meters). The optimal match is calculated using the Kuhn-Munkres algorithm (Kuhn, 1955) in a bipartite graph.

## B EXPERIMENT DETAILS

All algorithms were trained and tested on the NVIDIA A40 Data Center GPU.

### B.1 HYPERPARAMETERS

In order to keep our results as general as possible, we try to avoid hyperparameter tuning and choose to train all agents with the optimization values suggested by OpenAI. Table 3 contains a summary of all hyperparameters used and their meaning.

Table 3: Training hyperparameters used for RLD3 and DRL baselines.

| Hyperparameter | Value | Description |
|---|---|---|
| Optimizer | Adam | Scheme to update the parameters |
| Activator | GELU | Nonlinear activation function in neurons |
| Learning rate | 0.01 | Initial optimization learning rate for Adam optimizer |
| Update rate | 0.01 | Updating ratio in soft update |
| Episode number | 1000 | Number of virtual days in simulator |
| Episode length | 1200 | Number of steps in single episode |
| Exploration number | 100 | Number of episodes for pure exploration in early experiment |
| Batch size | 512 | Batch size during optimization |
| Replay buffer size | 1.2e6 | Number of states stored in the replay buffer |
| Steps per update | 60 | Optimization interval |
| Window length | 60 | The maximal length of lasting interaction |
| Order number | 500 | Number of orders in a single episode |
| Driver number | 50 | Number of driver |

### B.2 SIMULATOR RANDOMNESS

To ensure reproducibility, we first sample 5000 orders with random state 0 from the dataset as our training order set and sample 50 drivers with random state 1. In the training process, we randomly sample 500 orders from the training order set for each episode, using the episode index as the random seed. Additionally, we also utilize the episode index as the random seed for algorithm optimization.

## C SUPPLEMENTAL EXPERIMENTS

We provide additional experiments to support the challenges faced by various baselines in designated driver dispatching problems.

### C.1 INSTABILITY ISSUE CAUSED BY SPARSE FEEDBACK

As discussed in Sec 4.2, MADDPG and MAMFRL struggle to learn and differentiate in different directions when facing sparse feedback, especially in cold areas with few surrounding orders. This is because, in the absence of shared networks or training experiences among agents, individual agents receive insufficient feedback from the environment, which hinders their ability to grasp the distribution of unmatched orders. The same issue also arises in the ablation study with 50 agents.

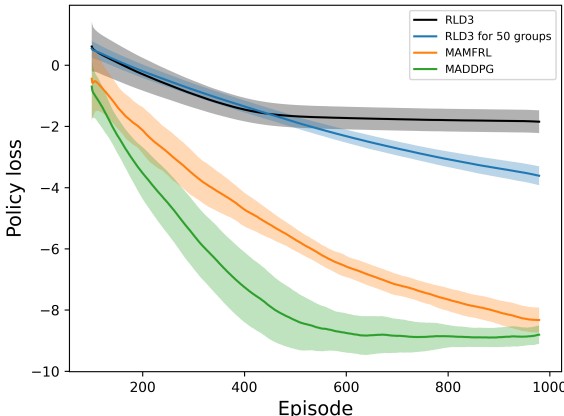

Figure 7: Policy loss in the training stage. The policy loss is the negative mean of the Q-values calculated from the data sampled from the buffer, since we update the actor-network using gradient descent as introduced in Equation (9). The order of the legends in the figure is the same as the order of performances in the last episode.

As shown in Figure 7, the convergence performance of policy loss in MADDPG, MAMFRL, and RLD3 for 50 groups is not as good as RLD3. Since the policy loss is the negative mean of the critic-network sampling, this indicates that traditional DRL algorithms without data sharing encounter the problem of Q-value overestimation when the experienced feedback for a single agent is sparse. Consequently, the objective of policy optimization becomes unstable, especially when facing states with few unmatched orders, making it difficult to differentiate between different directions of superiority or inferiority.

## C.2 OVER-EXPLORATION ISSUE CAUSED BY SPARSE DISTRIBUTION

It is worth noting that the intuition behind RND is to utilize an additional value function neural network to estimate intrinsic rewards, thereby encouraging the exploration of unknown state-action pairs and improving the performance of reinforcement learning algorithms in facing sparse feedback. However, in the context of designated driver dispatch problems, due to the slow movement of drivers and the significant moving cost involved, such unguided exploration of non-semantic information would result in agents having excessive moving distances.

As shown in Figure 8, in the early stages of training, due to the issue of over-exploration, almost all drivers in RND explore states that have not been visited before, resulting in high moving distances. However, in the later stages of training, due to the misleading effect of data with matching orders from distant locations, the drivers still fall into suboptimal solutions with long travel distances. In comparison, for RLD3 without the moving cost version, the drivers' distances are unrestricted, making it easy for them to repeatedly visit distant orders from historical data.

Figure 9 illustrates the early exploration process of agent #0 in MADDPG-RND, where multiple matches with distant orders located at the map boundaries lead to the learned strategy of always moving west. However, in reality, staying at the location and waiting for potential orders in the east would be more advantageous for agent #0.

Another interesting phenomenon is that in Table 1 of the main text, we tested the performance of the model with the optimal episodic reward during training. In this case, the optimal performance of MADDPG-RND occurs during the exploration phase in the early stages of training. Table 4 presents the comparison of the final trained models of MADDPG-RND, but its performance is still unsatisfactory.

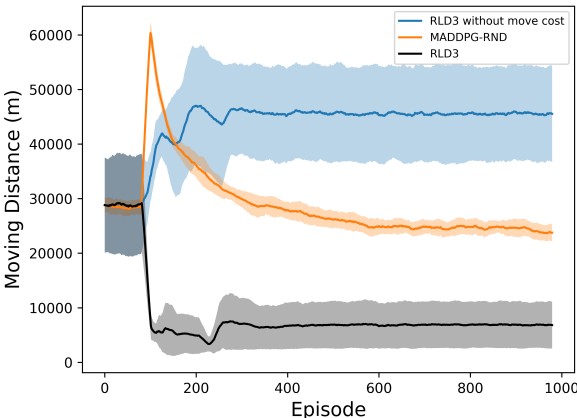

Figure 8: Moving distance in the training stage. The order of the legends in the figure is the same as the order of performances in the last episode.

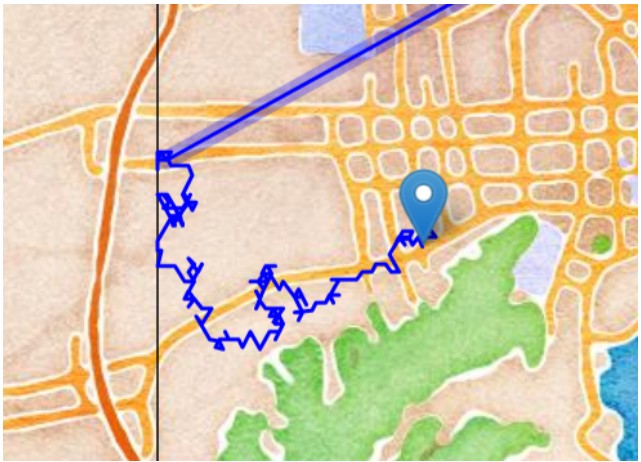

Figure 9: An example of misleading by exploration.

### C.3 DRIVER MOVING PATTERN HETEROGENEITY

We analyze drivers' behavior patterns and moving distances under moving constraints.

There are 5 groups in RLD3. Then the average moving distances of different cost scales are shown in Figure 10.

The result shows that with the increase in the move cost, the average moving distance of drivers decreases. However, it is interesting to note that the drivers' travel distances do not strictly decrease in accordance with increasing costs. Additionally, the relationship between travel distance and cost is not linear.

## D EXTRA FIGURES

### D.1 SIX-DIRECTION ACTION AND CORRESPONDING NEIGHBORHOOD SEGMENTATION

As introduced in Section 3.1, drivers' actions include seven discrete actions: stay, due East, North by 30 degrees east, North by 30 degrees west, due West, South by 30 degrees west, and South

Table 4: Extended Performance Comparison.

| Algorithm | Order | Distance (km) |
|---|---|---|
| RLD3 | $237.2 \pm 3.4$ | $7.0 \pm 1.3$ |
| MADDPG-RND (best episodic performance) | $228.6 \pm 3.5$ | $65.3 \pm 0.7$ |
| MADDPG-RND (final version) | $187.1 \pm 3.3$ | $23.8 \pm 0.5$ |

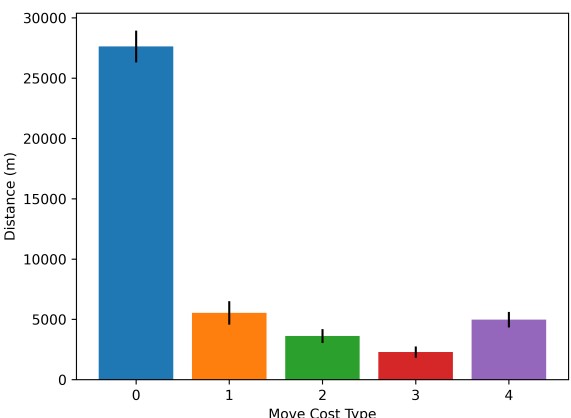

Figure 10: Move Distance for Different Groups.

by 30 degrees east, as shown in Figure 11. Then the space of the neighborhood of one driver is intuitively divided into six segments, each of which is extended by 30 degrees left and right in the corresponding action direction.

## D.2 DETAILED DESCRIPTION OF NEIGHBORHOOD REWARD

As introduced in Section 3.3, we use potential reward $nb_i^t$ to evaluate the potential for the location of agent $i$ to match nearby orders, so every driver at the same location has the same potential reward, which evaluates the distances and numbers of nearby unmatched orders and pays more attention on the closest order. As shown in Figure 12, the closest order contributes the majority of the potential value, while other orders in the neighborhood all contribute to the potential reward, and the closer the order, the greater the potential value. Such space-based reward encourages the driver to get closer to unmatched orders. As a result, drivers may have a smaller pick-up distance to nearby orders. Furthermore, as orders have 'hub-and-spoke' structure, drivers who are closer to locations with more orders are also more likely to receive future orders.

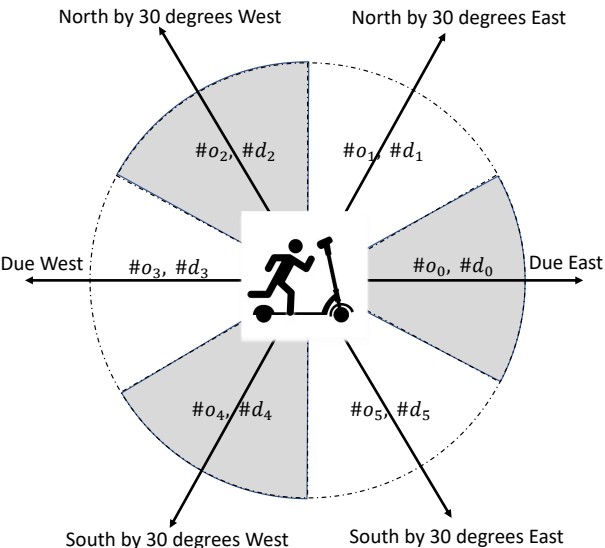

Figure 11: The six action directions and six segments of the neighborhood.

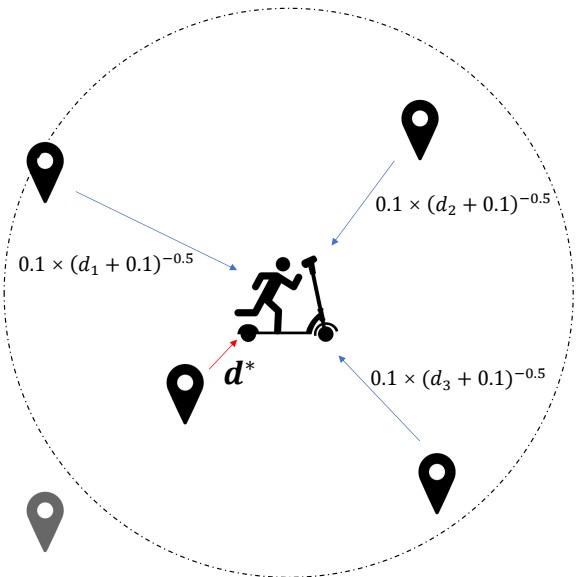

Figure 12: The neighborhood of a driver and the corresponding neighborhood reward $nb_i$.

