# OpenReview forum: "Sparsity-Aware Grouped Reinforcement Learning for Designated Driver Dispatch"
_ICLR.cc/2024/Conference — Submitted to ICLR 2024_

### Official Review · Reviewer_4mCM · 2023-10-23

**Soundness:** 3 good
**Presentation:** 2 fair
**Contribution:** 3 good
**Rating:** 6
**Confidence:** 4

**Summary:**

The paper proposed an algorithm (named RLD3) for solving Designated Driver Dispatch problem. The authors pointed out 3 challenges for this problem:
- data sparsity, i.e. number of designated drivers is considerably smaller than taxi drivers
- sparsity in reward, i.e. designated drivers only get reward once they are matched with an order
- difficulty in modeling cross-interaction effects, i.e. how driver A's behavior affects driver B, and their long-term impact on the agents' reward.
- hub-and-spoke issue, i.e. orders are concentrated in certain hotspots in the city, whereas the destinations are scattered. This results in long idle time of the driver after completing an order.

The paper framed the Designated Driver Dispatch problem as a decentralized partially observable markov decision process and proposed an actor-critic styled DRL solution. The authors proposed 3 ideas to tackle aforementioned challenges.

- Group sharing. This idea is to group agents into subgroups, members in the subgroup shares parameter weights (of the corresponding actor network) and experience
- Space potential. The idea is to mitigate the sparse reward issue by adding a prior to the reward based on the expected reward (computed from the available orders) in a region.
- Window lasting. The idea is to model agent-agent interaction by average actions over a time window. This is to mitigate the sparse and long term cross-interaction effect.

**Strengths:**

- formulation of the problem is reasonable
- the ideas for handling the challenges are reasonable and straightforward to implement

**Weaknesses:**

- evaluation metrics are completed orders and distance traveled, which is less direct than income earned
- the author mentioned the hub-and-spoke issue. but it not clear to me if any of the ideas proposed by the authors handles this issue
- presentation is not very clear (see follow-up questions below)

**Questions:**

1. Can you clarify the data sparsity issue, why 3000 drivers with 13000 orders a day is considered sparse. And what are the effect on the learning algorithm? The performance of other DRL algorithms are not too bad, which is not what I would expect for a data sparse environment.

2. Why use completed order and distance traveled as evaluation metrics? Income earned and distance traveled seem to be a more direct reward.

---

> ### Author Response · Authors · 2023-11-18
> **To Dear Reviewer 4mCM**
>
> Reviewer 4mCM:
>
> Thank you for providing your insightful comments on our paper. We genuinely appreciate the time and effort you have dedicated to reviewing our work. We have submitted a revised version of the paper with the newly added parts highlighted in red color. Below are our responses to the weaknesses and questions point-by-point:
>
> W1 & Q2: Evaluation metrics are completed orders and distance traveled, which is less direct than income earned.
>
> Answer to W1 & Q2: As described in Section 3.1, the choice of completed orders as an evaluation metric is motivated by the optimization of gross merchandise volume without using discriminatory order information. The increase in completed orders also means a better utilization of the supply, leading to a good overall income. Meanwhile, the completed order is also a fair and common metric that is perceivable by the public, including the potential passenger, driver, and investor, which could help the platform to gain an advantage in the market competition. [3, 4]
>
> Income is indeed an important objective, and we want to say that RLD3 can be readily adapted to price cases by changing the fix-value matching reward in Section 3.1 into the price of the order and incorporating order prices as weights of nearby orders' potentials in Section 3.3, without affecting the underlying grouping structure or agent interaction design.
>
> W2: The author mentioned the hub-and-spoke issue. but it not clear to me if any of the ideas proposed by the authors handles this issue.
>
> Answer to W2: We apologize for the unclear clarification of ‘hub-and-spoke’ structure and the corresponding sparsity issue. As introduced in Section 1, orders’ origins concentrate in hotspots while their destinations are dispersive, leading to the result that drivers are far from hot-spot. To address this challenge, we propose a ‘neighborhood reward’ to estimate the potential of a location. The ‘nb’ reward evaluates the distances and numbers of nearby unmatched orders and pays more attention to the closest order. Such space-based reward encourages the driver to get closer to the hot-spot. As a result, drivers may have a smaller pick-up distance to nearby orders. Furthermore, as orders have ‘hub-and-spoke’ structure, drivers who are closer to hotspots are also more likely to receive future orders. Additionally, since training data is shared by agents within the group, the agent's hot spot exploration efficiency is also improved.
>
> Q1: Can you clarify the data sparsity issue, why 3000 drivers with 13000 orders a day is considered sparse. And what are the effect on the learning algorithm? The performance of other DRL algorithms are not too bad, which is not what I would expect for a data sparse environment.
>
> Answer to Q1: We appreciate your request for clarification on the data sparsity issue. Hangzhou is a large city with 16850 km^2 area and 12 million population, so the scenario of 3000 drivers and 13000 orders is considered a sparse case compared with the taxi-dispatching scenario. Specifically, the average designated driver can only receive 3 to 5 orders per day, which is fundamentally different from the dozens of taxi orders per day. In addition, considering the distance between the drivers and the orders, the speed of the designated driver is far less than that of the taxi driver.
>
> However, 3000 agents would lead to scalability issues. Traditional CTDE algorithms often include the observation-action pair of all agents into the input of the value function to measure the impact of other agent policy changes on the environment, so that with the increase of the number of agents, the value function will face the problem of explosive input dimension and training efficiency [1, 2]. Thus, we sample the data set from a square with about 600 km^2 including 50 agents and 500 orders in our experiments. The experimental results and real data also show that except for several over-explored algorithms, the average distance of most designated drivers is about 25km, which is the side length of the area we selected. So we focus on the sub-area with a reasonable scale.
>
> As for baselines’ performance, it is worth noting that the market is very large as discussed in Section 1. So such an improvement can not only reduce the cost of the driver's movement, and improve the quality of service but also bring a huge improvement to the platform revenue.

---

> > ### Comment · Reviewer_4mCM · 2023-11-22
> >
> > Thanks for the clarification

---

> ### Author Response · Authors · 2023-11-18
> **To Dear Reviewer 4mCM (Reference)**
>
> Reference:
> [1] Huang, Wenhan and Li, Kai and Shao, Kun and Zhou, Tianze and Taylor, Matthew and Luo, Jun and Wang, Dongge and Mao, Hangyu and Hao, Jianye and Wang, Jun and Others,
> Multiagent q-learning with sub-team coordination,
> Advances in Neural Information Processing Systems, Volume 35, 2022.
>
> [2] Kai Cui, Anam Tahir, Gizem Ekinci, Ahmed Elshamanhory, Yannick Eich, Mengguang Li, Heinz Koeppl,
> A Survey on Large-Population Systems and Scalable Multi-Agent Reinforcement Learning,
> arxiv preprint arxiv:2209.03859, 2022.
>
> [3] Yang Liu, Fanyou Wu, Cheng Lyu, Shen Li, Jieping Ye, Xiaobo Qu,
> Deep dispatching: A deep reinforcement learning approach for vehicle dispatching on online ride-hailing platform,
> Transportation Research Part E: Logistics and Transportation Review, Volume 161, 2022.
>
> [4] T. Oda and C. Joe-Wong,
> MOVI: A Model-Free Approach to Dynamic Fleet Management,
> IEEE Conference on Computer Communications, Honolulu, HI, USA, 2018, pp. 2708-2716.

---

### Official Review · Reviewer_Y79N · 2023-10-29

**Soundness:** 2 fair
**Presentation:** 1 poor
**Contribution:** 2 fair
**Rating:** 5
**Confidence:** 4

**Summary:**

The paper proposes a new MARL algorithm for driver dispatch problems. Some tricks are applied for solving data, feedback and interaction sparsity to improve the performances. Simulation results show that the proposed method has superior performances over baseline methods including several MARL methods.

**Strengths:**

The paper investigates an interesting and new problem of driver dispatch. The proposed method is shown to be better than optimization baselines and other RL type algorithms.

**Weaknesses:**

My major critique is that although the paper demonstrates a usage of MARL algorithms on an interesting order dispatching problem, the overall method is quite straightforward with necessary engineering design choices to achieve the best performance. From a methodology view, the underlying techniques like mean-field with grouping, actor-critic, delayed update, etc exist in previous literature therefore lack originality.

Apart from above, the writing of the paper needs to be improved. Some parts are less clear to me and require improvement.

The usage of K-means on the encoder needs to be further explained.

For the training data, 50 drivers and 500 orders are a relatively small portion of the entire dataset, would it lead to biased policies? Why not train on a larger amount of data?

Some formulations are not clear:

In Eq. (8)(9), what is the sample to be expected over, and what’s the distribution of the sample?

Some sentences are confusing:

``we assign potential values to nearby unmatched orders, with higher feedback given to closer orders’’, is the potential value the same as feedback here?


``This encoder is responsible for the distribution of the current unmatched orders and idling agents respectively.’’



Minor issues:

Typo in definition of the observation in Sec. 3.1

**Questions:**

In the formulation of Dec-POMDP, the observation does not include the current statues of each driver (i.e., offline, idle or serving), how to ensure the actions for each drive is valid?

What are the six neighboring directions in the action space?

In Eq.(4), why is the cost of moving as a constant only depends on the group index?

How does the proposed algorithm compare with other single-agent RL algorithms for all drivers?

It seems the group number has a great effect on the performance. How is the group number of RLD3 selected? Does it depend on training data sample distributions?

How does MADDPG-RND and MAMFRL perform with grouping?

---

> ### Author Response · Authors · 2023-11-18
> **To Dear Reviewer Y79N**
>
> Thank you for your careful comments. We appreciate the time and effort you have put into reviewing our work. We have submitted a revised version of the paper with the newly added parts highlighted in red color. Below are our responses point-by-point:
>
> W1: My major critique is that although the paper demonstrates a usage of MARL algorithms on an interesting order dispatching problem, the overall method is quite straightforward with necessary engineering design choices to achieve the best performance. From a methodology view, the underlying techniques like mean-field with grouping, actor-critic, delayed update, etc exist in previous literature therefore lack originality.
>
> Answer to W1: To the best of our knowledge, designated driver dispatch is a new application scenario with few practical algorithms. Compared with the application of taxi dispatch, designated driver dispatch faces a unique sparsity challenge, which can be attributed to three factors: sparse dataset, sparse feedback, and sparse agent interaction.
>
> There exist similar works using group structure. However, they either focus on communication within groups [1, 3] or factorize the value function for scalability consideration [2], which does not address the sparsity issue in our scenario. In RLD3, the motivation behind grouping is to mitigate the sparsity of data. Within the group, agents share not only the strategy but also the data and training process, which can solve sparse data issues, i.e., by grouping agents of the same cost we have much denser data for agents of the same type (of cost). Furthermore, since we reduced the input dimension of the value function by group-sharing, RLD3 can handle a larger scale of agents.
>
> For agent interaction estimation techniques, traditional mean-field techniques treat the impact of all other agents’ policy changes as unitary feedback [4, 5]. However, agent interaction is complex in designated driver dispatch since drivers’ distribution is sparse and they move slowly. As introduced in Section 3.4, a driver’s income is not directly influenced by the one-step action of drivers located far away, but rather by the accumulated driver distribution changes caused by the lasting movements of drivers. Thus, RLD3 addresses the long-term agent interaction by including every group’s mean-field effect of window-lasting actions in the value function input to capture the inherent non-stationarity.
>
> Additionally, we also have a special reward design to estimate the potential of a location in matching with orders to fill in the gaps of sparse feedback. The ‘nb’ reward evaluates the distances and numbers of nearby unmatched orders and pays more attention to the closest order. Such space-based reward encourages the driver to get closer to unmatched orders. As a result, drivers may have a smaller pick-up distance to nearby orders. Furthermore, as orders have ‘hub-and-spoke’ structure, drivers who are closer to locations with more orders are also more likely to receive future orders.
>
> W2: The usage of K-means on the encoder needs to be further explained.
>
> Answer to W2: We apologize for the unclear clarification about the K-means encoder. The environment maintains a driver pool and an order pool to implement the state transfer. At each time step, we apply K-means on the coordinates of idling drivers $[(x_1, y_1), (x_2, y_2), …, (x_s, y_s)]$ where $s$ is the number of idling drivers, and get $K$ cluster centers $[(x^*_1, y^*_1), (x^*_2, y^*_2), …, (x^*_K, y^*_K)]$ as the encoded hot-spot distribution of idling drivers. Similarly, we can get the $K$ cluster centers for unmatched orders. Then we use the concatenation of the two sets of centers as the encoded vector for state $s$. Such an encoder is able to handle a changing number of drivers and orders while representing hot spots in the map.
>
> W3: For the training data, 50 drivers and 500 orders are a relatively small portion of the entire dataset, would it lead to biased policies? Why not train on a larger amount of data?
>
> Answer to W3: Our 3000-driver and 13000-order data come from Hangzhou, a large city with a 16850 km^2 area and a 12 million population. However, 3000 agents are too large for MARL in practice. Traditional CTDE algorithms often include the observation-action pair of all agents into the input of the value function to measure the impact of other agent policy changes on the environment, so that with the increase of the number of agents, the value function will face the problem of explosive input dimension and training efficiency [6,7]. Thus, we sample the data set from a square with about 600 km^2 including 50 agents and 500 orders in our experiments. The experimental results and real data also show that except for several over-explored algorithms, the average distance of most designated drivers is about 25km, which is the side length of the area we selected. So we focus on the sub-area with a reasonable scale.

---

> ### Author Response · Authors · 2023-11-18
> **To Dear Reviewer Y79N (continued)**
>
> W4: In Eq. (8)(9), what is the sample to be expected over, and what’s the distribution of the sample?
>
> Answer to W4: The expectation is over the whole experience buffer of agent
>  $i$ at time step $t$ as we only sample a batch of data from the whole experience buffer.
>
> W5: ``we assign potential values to nearby unmatched orders, with higher feedback given to closer orders’’, is the potential value the same as feedback here?
>
> Answer to W5: We apologize for the ambiguous representation. It is the same, we use potential value (d+0.1)^{-0.5} with weights as feedback to the agent.
>
> W6: ``This encoder is responsible for the distribution of the current unmatched orders and idling agents respectively.’’
>
> Answer to W6: We apologize again for the unclear representation. As explained in answer to W2, we use K-means as the encoder to represent hot spots of orders and distribution of drivers.
>
> Q1: In the formulation of Dec-POMDP, the observation does not include the current statues of each driver (i.e., offline, idle or serving), how to ensure the actions for each drive is valid?
>
> Answer to Q1: As introduced in the formulation in Section 3.1, the observation vector is the input of the driver's policy function. In the simulator, we maintain the status of the driver in the state to ensure a valid action. We have added images in Appendix A to provide a more detailed illustration of the simulator implementation for state transfer of drivers and orders.
>
> Q2: What are the six neighboring directions in the action space?
>
> Answer to Q2: As illustrated in Figure 1, the space is covered by a hexagon partition, so the six action directions include due East, North by 30 degrees east, North by 30 degrees west, due West, South by 30 degrees west, and South by 30 degrees east. We have added an image in Appendix D.1 to enhance the understanding.
>
> Q3: In Eq.(4), why is the cost of moving as a constant only depends on the group index?
>
> Answer to Q3: We apologize for the confusing presentation about grouping structure. The motivation behind grouping is to reflect the endogenous heterogeneity of agents, particularly in the context of the cost types of drivers. Within the group, as agents have the same type, the group shares not only the strategy but also the data and training process, which can solve sparse data issues. Furthermore, since we reduced the input dimension of the value function by group-sharing, the algorithm should not have the scalability issue of traditional CTDE when fixing the group number.
>
> Q4: How does the proposed algorithm compare with other single-agent RL algorithms for all drivers?
>
> Answer to Q4: As we mentioned before, joint-policy single-agent RL has scalability issues due to high-dimension action space and value function input. On the other hand, sharing-policy single-agent RL lacks coordination of multi-agent interactions since changes in other agents' policies can lead to non-stationary in the environment. We did notice an interesting previous work, deep-dispatch, in which individual local information and global information are combined to help with sharing single-agent RL in taxi-dispatch [8]. As shown in Table 1, RLD3 still outperforms deep-dispatch.
>
> Q5: It seems the group number has a great effect on the performance. How is the group number of RLD3 selected? Does it depend on training data sample distributions?
>
> Answer to Q5: As we mentioned before, the choice of the group number comes from the heterogeneity of agents’ cost type. In the field of user modeling, agent type 3-5 is often enough to model the attitude tendency of different agents [9, 10].
>
> Additionally, as shown in Section 4.4, the ablation study on group number provides evidence of the efficacy of group-sharing. A larger group number can better represent the heterogeneity of drivers but would lead to sparser training data per agent. It’s shown that the group-sharing structure does help with the performance either for RLD3 or for MADDPG.
>
> Q6: How does MADDPG-RND and MAMFRL perform with grouping?
>
> Answer to Q6: We have run extra experiments on MADDPG-RND and MAMFRL with grouping. As shown in Table 2 in the revised version, the group-sharing structure can help improve the performance of MAMFRL as the data sparsity issue is addressed. However, Since RND itself suffers from over-exploration, group structure does not help its performance.

---

> > ### Comment · Reviewer_Y79N · 2023-11-22
> >
> > Thanks for the detailed response.
> >
> > For additional method deep-dispatch as a single-agent RL approach, the performance is close to RLD3 but worse. From an information theoretical viewpoint, why does this method work worse than RLD3? Is there any lack of information or errors in approximation leading to its worse performance? Otherwise it can be mostly affected by the size of the models, hyperparameter tuning, etc. Analysis on this can be helpful.
> >
> > MAMFRL with grouping seems to be better than RLD3 in distance, this makes RLD3 less superior.
> >
> > Also I would recommend to add more details in the main paper based on above response.
> >
> > Based on above, I raise my score but still with concerns about the novelty of the method and its scalability.

---

> > > ### Author Response · Authors · 2023-11-23
> > >
> > > Thank you for taking the time to consider our response and for providing further feedback.
> > >
> > > Regarding your point about the performance comparison between deep-dispatch and RLD3, it is important to note that deep-dispatch combines the driver's local and global state information in the policy input, which gives deep-dispatch an inherent information advantage over decentralized strategies that take observations as input. Since deep-dispatch puts the data of different drivers on a single agent to learn from, it addresses the data sparsity issue to some extent. However, since the deep-dispatch single-agent does not deal with the heterogeneity of drivers of different cost types, its unified estimation of different driver policies may cause the algorithm to fall into the suboptimal position, even if the global information is utilized in the unified policy. Additionally, deep-dispatch does not estimate time-lasting agent interaction, so the coordination among multi-agents is not as good as RLD3. The ablation study of RLD3 without window-lasting and RLD3 for 50 groups shows the importance of agent heterogeneity and agent interaction issues.
> > >
> > > Regarding the performance of MAMFRL with grouping in distance metrics compared to RLD3, although it achieves less distance than RLD3, its normalized reward is still inferior to RLD3 when combined with the difference in completed orders. The good performance of MAMFRL with 5 groups also shows that our proposed group-sharing structure can improve the performance of other MARL algorithms.
> > >
> > > We are grateful for the opportunity to improve our work based on your valuable comments. We will address your concerns about the comparison between former works and RLD3 in the future version.

---

> ### Author Response · Authors · 2023-11-18
> **To Dear Reviewer Y79N (reference)**
>
> Reference:
> [1] Shao, J., Lou, Z., Zhang, H., Jiang, Y., He, S., & Ji, X,
> Self-Organized Group for Cooperative Multi-agent Reinforcement Learning,
> Advances in Neural Information Processing Systems, 2022.
>
> [2] Rashid, T., Samvelyan, M., De Witt, C. S., Farquhar, G., Foerster, J., & Whiteson, S,
> Monotonic value function factorisation for deep multi-agent reinforcement learning,
> The Journal of Machine Learning Research, 2020, 21(1), 7234-7284.
>
> [3] Meng X, Tan Y.
> Learning Group-Level Information Integration in Multi-Agent Communication,
> Proceedings of the 2023 International Conference on Autonomous Agents and Multiagent Systems, 2023.
>
> [4] Sriram Ganapathi Subramanian, Pascal Poupart, Matthew E. Taylor, and Nidhi Hegde,
> Multi type mean field reinforcement learning,
> In Proceedings of the 19th International Conference on Autonomous Agents and Multi-Agent Systems, pp. 411–419, Richland, SC, 2020.
>
> [5] Yaodong Yang, Rui Luo, Minne Li, Ming Zhou, Weinan Zhang, and Jun Wang,
> Mean field multi-agent reinforcement learning,
> In Proceedings of the 35th International Conference on Machine Learning, July 10-15, 2018.
>
> [6] Huang, Wenhan and Li, Kai and Shao, Kun and Zhou, Tianze and Taylor, Matthew and Luo, Jun and Wang, Dongge and Mao, Hangyu and Hao, Jianye and Wang, Jun and Others,
> Multiagent q-learning with sub-team coordination,
> Advances in Neural Information Processing Systems, Volume 35, 2022.
>
> [7] Kai Cui, Anam Tahir, Gizem Ekinci, Ahmed Elshamanhory, Yannick Eich, Mengguang Li, Heinz Koeppl,
> A Survey on Large-Population Systems and Scalable Multi-Agent Reinforcement Learning,
> arxiv preprint arxiv:2209.03859, 2022.
>
> [8] Yang Liu, Fanyou Wu, Cheng Lyu, Shen Li, Jieping Ye, Xiaobo Qu,
> Deep dispatching: A deep reinforcement learning approach for vehicle dispatching on online ride-hailing platform,
> Transportation Research Part E: Logistics and Transportation Review, Volume 161, 2022.
>
> [9] Mohammad Shahverdy, Mahmood Fathy, Reza Berangi, Mohammad Sabokrou,
> Driver behavior detection and classification using deep convolutional neural networks,
> Expert Systems with Applications, Volume 149, 2020.
>
> [10] María J. Alonso-González, Sascha Hoogendoorn-Lanser, Niels van Oort, Oded Cats, Serge Hoogendoorn,
> Drivers and barriers in adopting Mobility as a Service (MaaS) – A latent class cluster analysis of attitudes,
> Transportation Research Part A: Policy and Practice, Volume 132, 2020, Pages 378-401.

---

### Official Review · Reviewer_z3iz · 2023-10-31

**Soundness:** 3 good
**Presentation:** 2 fair
**Contribution:** 2 fair
**Rating:** 5
**Confidence:** 4

**Summary:**

This research introduces Reinforcement Learning for Designated Driver Dispatch (RLD3) to address driver dispatch challenges in the designated driving market. The algorithm leverages group-sharing structures and diverse rewards, achieving a balance between heterogeneity, sparsity, and scalability. The model also accounts for long-term agent interactions and is evaluated using real-world data.

**Strengths:**

1. The proposed algorithm, Reinforcement Learning for Designated Driver Dispatch (RLD3), introduces innovative features like group-sharing structures and window-lasting policy ensembles, highlighting its novelty and potential effectiveness.
2. The study innovatively leverages group-sharing structures in reinforcement learning to address dataset sparsity, ensuring both heterogeneity and scalability in driver dispatch.
3. By considering both short-term rewards with heterogeneous costs and long-term agent cross-effects, the proposed algorithm promises a holistic approach to the driver dispatch problem.

**Weaknesses:**

1. Merely introducing group-sharing structures does not necessarily guarantee improved results in reinforcement learning; evidence of its efficacy is lacking.
2. It's concerning that the study does not compare RLD3 with state-of-the-art algorithms in designated driver dispatch, rendering its so-called "superior performance" questionable.
3. The paper's vague assertions, such as achieving a trade-off between heterogeneity and scalability, are made without concrete empirical evidence, undermining the study's credibility.

**Questions:**

1. How does the proposed group-sharing structure in RLD3 compare to other multi-agent coordination mechanisms prevalent in multi-agent reinforcement learning? The choice of group-sharing seems arbitrary without a clear benchmark or rationale.
2. In real-world scenarios like driver dispatch, agent-agent interactions can be complex. How does RLD3 handle potential negative emergent behaviors between agents, especially when considering group-shared policies?
3. It's mentioned that the algorithm addresses long-term agent cross-effects. However, in multi-agent systems, understanding and addressing non-stationarity is crucial. How does RLD3 handle the inherent non-stationarity in multi-agent environments?
4. Given the heterogeneity of agents in your setup, how does RLD3 ensure fair allocation of rewards and prevent potential domination by certain groups or agents, which could lead to suboptimal overall system performance?

**Details Of Ethics Concerns:**

No.

---

> ### Author Response · Authors · 2023-11-18
> **To Dear Reviewer z3iz**
>
> Thank you for providing detailed feedback on our paper. We appreciate the time and effort you have spent in reviewing our work. We have submitted a revised version of the paper with the newly added parts highlighted in red color. Below are our responses to the weaknesses and questions point-by-point:
>
> W1: Merely introducing group-sharing structures does not necessarily guarantee improved results in reinforcement learning; evidence of its efficacy is lacking.
>
> Answer to W1: We apologize for the obscure presentation about grouping structure. The motivation behind grouping is to reflect the endogenous heterogeneity of agents, particularly in the context of the cost types of drivers. We group the agents that have the same cost together and let them share the same policy, as well as the data and training process, to mitigate the sparsity of data. Furthermore, since we reduced the input dimension of the value function by group-sharing, the algorithm can handle a larger scale of agents.
>
> As shown in Section 4.4, the ablation study on group number provides evidence of the efficacy of group-sharing. A larger group number can better represent the heterogeneity of drivers but would lead to sparser training data per agent. It’s shown that the group-sharing structure does help with the performance either for RLD3 or for MADDPG.
>
> We have run extra experiments on MADDPG-RND and MAMFRL with grouping. As shown in Table 2 in the revised version, the group-sharing structure can help improve the performance of MAMFRL as the data sparsity issue is addressed. However, Since RND itself suffers from over-exploration, group structure does not help its performance.
>
> W2: It's concerning that the study does not compare RLD3 with state-of-the-art algorithms in designated driver dispatch, rendering its so-called "superior performance" questionable.
>
> Answer to W2: To the best of our knowledge, designated driver dispatch is a new application scenario with few practical algorithms. Compared with the application of taxi dispatch, designated driver dispatch faces the unique sparsity challenge, which can be attributed to three factors: sparse dataset, sparse feedback, and sparse agent interaction. Thus, we compare RLD3 with other popular MARL algorithms that address sparsity issues or scalability issues.
>
> In the revised manuscript, we have added a comparison between RLD3 and deep-dispatch, the state-of-the-art algorithm in the taxi-dispatch problem [1]. As shown in Table 1, RLD3 still outperforms deep-dispatch.
>
> W3: The paper's vague assertions, such as achieving a trade-off between heterogeneity and scalability, are made without concrete empirical evidence, undermining the study's credibility.
>
> Answer to W3: For scalability, as shown in Section 4.4, we have run an ablation study on the group number. We can see that without group-sharing structure, most CTDE MARL algorithms are facing scalability issues, such as MADDPG, and RLD3 with 50 groups (which means one agent in each group). Such CTDE algorithms often include the observation-action pair of all agents into the input of the value function to measure the impact of other agent policy changes on the environment, so that with the increase of the number of agents, the value function will face the problem of explosive input dimension and training efficiency [2]. RLD3 reduces the input dimension of the value function by group-sharing, so RLD3 can handle a larger scale of agents.
>
> For heterogeneity, as clarified in the response to W1, we divide the drivers into groups according to the endogenous heterogeneity of drivers. We have also referenced previous works that support the effectiveness of dividing into classes when facing heterogeneity [3]. In our experiment, there are 5 types of drivers. However, in practical application, the number of types (groups) can be generalized to other different settings.

---

> ### Author Response · Authors · 2023-11-18
> **To Dear Reviewer z3iz (continued)**
>
> Q1: How does the proposed group-sharing structure in RLD3 compare to other multi-agent coordination mechanisms prevalent in multi-agent reinforcement learning? The choice of group-sharing seems arbitrary without a clear benchmark or rationale.
>
> Answer to Q1: The existing CTDE frameworks often include the observation-action pair of all agents into the input of the value function to measure the impact of other agent policy changes on the environment, so that with the increase of the number of agents, the value function will face the problem of explosive input dimension and training efficiency [2, 4]. Additionally, traditional mean-field techniques treat the impact of all other agents’ policy changes as unitary feedback [8, 9]. However, we coordinate the impact of other agents on the environment by adding the mean-field effect of every group in the value function as shown in Figure 3. We also use window-lasting action to address the long-term interaction.
>
> There exist similar works using group structure to address heterogeneity. However, they either focus on communication within groups [5, 7], or factorize the value function for scalability consideration [6], which do not address the sparsity issue in our scenario. In RLD3, the motivation behind grouping is to mitigate the sparsity of data. Within the group, agents share not only the strategy but also the data and training process, which can solve sparse data issues, i.e., by grouping agents of the same cost we have much denser data for agents of the same type (of cost). Furthermore, since we reduced the input dimension of the value function by group-sharing, so RLD3 can handle a larger scale of agents.
>
> Q2: In real-world scenarios like driver dispatch, agent-agent interactions can be complex. How does RLD3 handle potential negative emergent behaviors between agents, especially when considering group-shared policies?
>
> Answer to Q2: For agent coordination, on the one hand, we avoid the subjective influence of agents among different groups, leaving only the endogenous type differences. On the other hand, as introduced in Section 3.2, Gumbel-softmax policy ensures that agents can learn strategies in a mixed strategy space. When the learning of value function converges, the policy space can cover Mixed Strategy Nash Equilibrium. Therefore, even if negative emergent behavior occurs early in the training, the algorithm can be adjusted adaptively through the reward. For example, if there are two drivers competing for the same order, the reward of one driver is reduced, and the driver will later prefer the behavior with higher rewards if he explores other behaviors with higher rewards.
>
> Q3: It's mentioned that the algorithm addresses long-term agent cross-effects. However, in multi-agent systems, understanding and addressing non-stationarity is crucial. How does RLD3 handle the inherent non-stationarity in multi-agent environments?
>
> Answer to Q3: We apologize for the unclear clarification of long-term agent interaction. As introduced in Section 3.4, a driver’s income is not directly influenced by the one-step action of drivers located far away, but rather by the accumulated driver distribution changes caused by the lasting movements of drivers. RLD3 addresses the long-term agent interaction by including the mean-field effect of window-lasting actions in the value function input to capture the inherent non-stationarity. The distance between drivers is far considering their moving speed, and the driver’s observation includes the distribution of nearby drivers. Therefore, the estimation of the movement trend of drivers of different types is sufficient to reflect the destabilizing effects of other agent's policy changes on the environment.
>
> Q4: Given the heterogeneity of agents in your setup, how does RLD3 ensure fair allocation of rewards and prevent potential domination by certain groups or agents, which could lead to suboptimal overall system performance?
>
> Answer to Q4: As introduced in Section 3.1, individual reward is considered in our research. So RLD3 does not involve reward allocation but optimizes fair individual reward. We also analyze the agent behavior pattern of different groups to show that there is no domination issue, the only difference is the moving cost type.

---

> ### Author Response · Authors · 2023-11-18
> **To Dear Reviewer z3iz (Reference)**
>
> Reference:
> [1] Yang Liu, Fanyou Wu, Cheng Lyu, Shen Li, Jieping Ye, Xiaobo Qu,
> Deep dispatching: A deep reinforcement learning approach for vehicle dispatching on online ride-hailing platform,
> Transportation Research Part E: Logistics and Transportation Review, Volume 161, 2022.
>
> [2] Huang, Wenhan and Li, Kai and Shao, Kun and Zhou, Tianze and Taylor, Matthew and Luo, Jun and Wang, Dongge and Mao, Hangyu and Hao, Jianye and Wang, Jun and Others,
> Multiagent q-learning with sub-team coordination,
> Advances in Neural Information Processing Systems, Volume 35, 2022.
>
> [3] Mondal W U, Agarwal M, Aggarwal V, et al. ,
> On the approximation of cooperative heterogeneous multi-agent reinforcement learning (marl) using mean field control (mfc),
> The Journal of Machine Learning Research, 2022, 23(1): 5614-5659.
>
> [4] Kai Cui, Anam Tahir, Gizem Ekinci, Ahmed Elshamanhory, Yannick Eich, Mengguang Li, Heinz Koeppl,
> A Survey on Large-Population Systems and Scalable Multi-Agent Reinforcement Learning,
> arxiv preprint arxiv:2209.03859, 2022.
>
> [5] Shao, J., Lou, Z., Zhang, H., Jiang, Y., He, S., & Ji, X,
> Self-Organized Group for Cooperative Multi-agent Reinforcement Learning,
> Advances in Neural Information Processing Systems, 2022.
>
> [6] Rashid, T., Samvelyan, M., De Witt, C. S., Farquhar, G., Foerster, J., & Whiteson, S,
> Monotonic value function factorization for deep multi-agent reinforcement learning,
> The Journal of Machine Learning Research, 2020, 21(1), 7234-7284.
>
> [7] Meng X, Tan Y.
> Learning Group-Level Information Integration in Multi-Agent Communication,
> Proceedings of the 2023 International Conference on Autonomous Agents and Multiagent Systems, 2023.
>
> [8] Sriram Ganapathi Subramanian, Pascal Poupart, Matthew E. Taylor, and Nidhi Hegde,
> Multi type mean field reinforcement learning,
> In Proceedings of the 19th International Conference on Autonomous Agents and Multi-Agent Systems, pp. 411–419, Richland, SC, 2020.
>
> [9] Yaodong Yang, Rui Luo, Minne Li, Ming Zhou, Weinan Zhang, and Jun Wang,
> Mean field multi-agent reinforcement learning,
> In Proceedings of the 35th International Conference on Machine Learning, July 10-15, 2018.

---

### Official Review · Reviewer_A8WA · 2023-10-31

**Soundness:** 3 good
**Presentation:** 1 poor
**Contribution:** 2 fair
**Rating:** 5
**Confidence:** 3

**Summary:**

Designated driver services are rapidly growing businesses.
The research challenge for this business is determining how to design the dispatch of designated drivers.
To address this issue, we propose RLD3, with the following key elements:
1. A structure where groups share experiences.
2. A reward function that considers trade-offs among heterogeneity, sparsity, and scalability.
3. A Window-lasting ensemble policy for long-term rewards.
We implemented and evaluated the above in a simulation environment, and it outperformed other deep reinforcement learning and optimization methods.

**Strengths:**

1. The description of the research background addressed in the paper is clear.

2. The approach of directing the movement of the designated driver is innovative.

3. The explanations for N, S, O, A, P, R, and γ in the formulation are clear.

**Weaknesses:**

1. The simulation environment used for training and evaluation is not well-illustrated in the text. It would be helpful if diagrams or images were included for a more detailed understanding.

2. I don't quite understand the reward function 'nb' that was established for the 'space potential'. A more detailed explanation is needed.

**Questions:**

1. "You have set the time step to 30 minutes. Is it appropriate for the agent to perform the same action for 30 minutes? For example, if you move in one direction with an electric scooter for 30 minutes, you'll travel several kilometers, which I expect would take you past the Hub area. Could you visualize how the agent acted over time in the simulation environment?"

2. "When looking at the reward graph by episode, most RL Methods show a sharp increase in rewards around the 100th iteration, with little difference afterward. This is different from typical learning graphs. Why do you think the learning occurred in this manner?"

3. In the Observation, it's mentioned as a 22-dimensional vector, but how is the #order and #driver composed into a 6-dimensional vector? Also, what is the "Six-segment-direction"?

---

> ### Author Response · Authors · 2023-11-18
> **To Dear Reviewer A8WA**
>
> Thank you for your careful comments. We appreciate the time and effort you have put into reviewing our work. We have submitted a revised version of the paper with the newly added parts highlighted in red color. Below are our responses point-by-point:
>
> W1: The simulation environment used for training and evaluation is not well-illustrated in the text. It would be helpful if diagrams or images were included for a more detailed understanding.
>
> Answer to W1: We acknowledge the need for a more detailed illustration of the simulation environment used for training and evaluation. As per your suggestion, we have added images in Appendix A to provide a more detailed introduction about the simulator implementation, including the state transfer of drivers and orders.
>
> For the order process, every new-coming order enters the order pool and waits to be matched. During the waiting period, if an unmatched order is not answered within a specified period (15 minutes), it enters the overtime status. Additionally, each order may be canceled via a Poisson Process with a mean patience of 8 minutes or real-world time. When an unmatched order is matched with a driver, it enters the on-service state and then transitions to the completion state after the expected completion time.
>
> Each driver has a scheduled on-work and off-work time. Idling drivers can either move to a given location according to the dispatch policy or match with an order for service. When an idling driver is matched with an order, the driver immediately moves to pick up the customer. Once the order is completed, the driver returns to the idle status until the simulation time exceeds their offline time.
>
> W2: I don't quite understand the reward function 'nb' that was established for the 'space potential'. A more detailed explanation is needed.
>
> Answer to W2: We apologize for the typo in the definition of ‘nb’, and we have corrected it in the revised version. We also add an image in Appendix D.2 to illustrate how ‘nb’ reward consists of. As introduced in Section 3.3, we use ‘nb’ to evaluate the potential for a location to match nearby orders, so every driver at the same location has the same ‘nb’ reward. The ‘nb’ reward evaluates the distances and numbers of nearby unmatched orders and pays more attention to the closest order. Such space-based reward encourages the driver to get closer to unmatched orders. As a result, drivers may have smaller pick-up distances to nearby orders. Furthermore, as orders have ‘hub-and-spoke’ structure, drivers who are closer to locations with more orders are also more likely to receive future orders.
>
> Q1: You have set the time step to 30 minutes. Is it appropriate for the agent to perform the same action for 30 minutes? For example, if you move in one direction with an electric scooter for 30 minutes, you'll travel several kilometers, which I expect would take you past the Hub area. Could you visualize how the agent acted over time in the simulation environment?
>
> Answer to Q1: We want to clarify that the time step is set to 30 seconds, as introduced in the second paragraph of Sec 3.1. With this time interval, a driver with a speed of 25 km/h is about to travel across one block (around 200m/650 feet).
>
> Q2: When looking at the reward graph by episode, most RL Methods show a sharp increase in rewards around the 100th iteration, with little difference afterward. This is different from typical learning graphs. Why do you think the learning occurred in this manner?
>
> Answer to Q2: We apologize for the confusing presentation, though we showed 100 pure exploration episode number in Table 3, we didn't clarify this until Appendix B.1. This is due to the 100-step pure exploration phase. We use the first 100 episodes for random pure exploration to mitigate the sparsity issue, which is highlighted in the revised version. It means that the policies of all drivers are set to be random in the first 100 episodes. After the 100th episode, we update the parameter of the policy every 60 steps, leading to about 20 updates in one episode on average.
>
> Q3: In the Observation, it's mentioned as a 22-dimensional vector, but how is the #order and #driver composed into a 6-dimensional vector? Also, what is the "Six-segment-direction"?
>
> Answer to Q3: We apologize for the confusion regarding the 22-dimensional observation vector and the composition of the 6-dimensional vector related to #order and #driver. As illustrated in Figure 1, the space is partitioned into hexagons, so the six action directions include due East, North by 30 degrees east, North by 30 degrees west, due West, South by 30 degrees west, and South by 30 degrees east. Therefore, we can divide the space into six segments, each of which is extended by 30 degrees left and right in the corresponding action direction. Thus, we can count the number of orders and drivers in each segment respectively. We have added an image in Appendix D.1 to enhance the understanding of these concepts.

---

### Meta-Review · Area_Chair_jgUx · 2023-12-06

**Metareview:**

This paper addresses the problem of driver dispatch for the designated driving service. It proposes new group-sharing structures to address dataset sparsity and window-lasting policy ensembles for sparse and lasting multi-agent interactions. The results show improvement over other multi-agent reinforcement learning methods.

The technical novelty and contribution of this paper are limited. The techniques underlying the proposed method exist in previous studies, making the contribution marginally. Moreover, designated driver dispatch is a large-scale problem. Currently, in the experiments, there are only 50 agents, which makes the proposed method inconclusive in solving real problems. Especially, this is an application-oriented paper. These are the main weaknesses but are not addressed during rebuttal.

**Justification For Why Not Higher Score:**

The technical contribution is limited, and how the proposed method can solve the real problem of driver dispatch is inconclusive due to the scalability concern of the current experiments.

**Justification For Why Not Lower Score:**

N/A

---

### Decision · Program_Chairs · 2024-01-16

Reject